# Comment on 'YcgC represents a new protein deacetylase family in prokaryotes'

**Magdalena Kremer[1†], Nora Kuhlmann[1†], Marius Lechner[1], Linda Baldus[1], Michael Lammers[1,2]\***

[1]Institute for Genetics and Cologne Excellence Cluster on Cellular Stress Responses in Aging-Associated Diseases (CECAD), University of Cologne, Cologne, Germany; [2]Institute of Biochemistry, Synthetic and Structural Biochemistry, University of Greifswald, Greifswald, Germany

**Abstract** Lysine acetylation is a post-translational modification that is conserved from bacteria to humans. It is catalysed by the activities of lysine acetyltransferases, which use acetyl-CoA as the acetyl-donor molecule, and lysine deacetylases, which remove the acetyl moiety. Recently, it was reported that YcgC represents a new prokaryotic deacetylase family with no apparent homologies to existing deacetylases (Tu et al., 2015). Here we report the results of experiments which demonstrate that YcgC is not a deacetylase.
DOI: https://doi.org/10.7554/eLife.37798.001

**\*For correspondence:**
michael.lammers@uni-greifswald.de

[†]These authors contributed equally to this work

**Competing interests:** The authors declare that no competing interests exist.

## Introduction

The progress in quantitative mass spectrometry in the last decade enabled the identification of thousands of lysine acetylation sites in all kingdoms of life (*Choudhary et al., 2009*; *Lundby et al., 2012*; *Weinert et al., 2014*, *2017*, *2013b*; *Zhang et al., 2009*). While histones were known since the 1960 s to be modified by lysine acetylation, these studies revealed that proteins of all cellular compartments covering essential cellular functions are lysine acetylated in eukaryotes, archaea and prokaryotes (*Finkemeier et al., 2011*; *König et al., 2014*). Lysine acetylation is a dynamic post-translational modification that is catalysed by lysine acetyltransferases, which add the acetyl moiety and lysine deacetylases (KDACs) which remove it. Site-specific non-enzymatic acetylation has also been reported (*Baeza et al., 2015*). While two classes of deacetylases exist in eukaryotes, the classical $Zn^{2+}$-dependent KDACs and the $NAD^+$-dependent sirtuins, prokaryotes only encode for sirtuins. *Escherichia coli* encodes for only one single KDAC, the sirtuin deacetylase CobB, which is structurally highly similar to eukaryotic sirtuins showing only minor structural differences in the small $Zn^{2+}$-binding domain (*Zhao et al., 2004*). In contrast, in mammals there are seven sirtuin deacylases (Sirt1-7), which show distinct subcellular localisation. While the nuclear Sirt1, the cytosolic Sirt2 and the mitochondrial Sirt3 possess a robust deacetylase activity, the remaining deacetylases show preferences for longer acylations or act as ADP-ribosyltransferases (*Du et al., 2011*; *Feldman et al., 2013*; *Liszt et al., 2005*; *Smith et al., 2008*). Mammals encode eleven classical KDACs and seven sirtuins enabling to use some activities with rather low specificity, while other enzymes show a remarkable specificity for some substrates and some lysine acylation sites (*Knyphausen et al., 2016*). This allows mammals to use post-translational lysine acylation as a molecular switch with consequences on signal transduction and regulation of protein function. In contrast, the prokaryotic CobB has a broad substrate range and it is quite promiscuous with respect to its substrate proteins. This suggests CobB's major role to be the detoxification of lysine acylations occuring upon metabolic fuel switching or metabolic stress under conditions of accumulation of acetyl-CoA or acetyl-phosphate, which can both serve as acetyl group donor molecules in prokaryotes (*Weinert et al., 2013a*, *2017*).

Recently, Tu et al. reported the presence of another deacetylase class in prokaryotes with YcgC as a representative (*Tu et al., 2015*). YcgC (also termed DhaM) together with YcgT (DhaK) and YcgS (DhaL) was originally reported to be part of the active dihydroxyacetone kinase (DhaK) complex. It is a component of a phosphorelay system using phosphoenolpyruvate as phosphoryl donor in which the phosphoryl group is finally transferred by DhaK to dihydroxyacetone yielding dihydroxyacetone phosphate (*Gutknecht et al., 2001*). YcgC has no sequence or structural homology to either eukaryotic classical KDACs or to sirtuin deacylases and it does not depend on $Zn^{2+}$ or $NAD^+$ for catalysis. Tu et al. reported that YcgC acts as serine hydrolase using a distinct catalytic mechanism involving a catalytic serine residue (S200); they also reported that YcgC catalyses deacetylation of the transcriptional regulator RutR and supports subsequent autoproteolytic cleavage at the RutR N-terminus (*Tu et al., 2015*). Here we report the results of our efforts to build on this work. In summary, we were unable to detect any deacetylase activity of YcgC, which means that CobB would seem to be the only deacetylase present in *E. coli*, and that YcgC should not be annotated as a lysine deacetylase in UniProt and other databases.

## Results

### Expression and purification of YcgC, CobB and RutR

We recombinantly expressed YcgC, the supposed deacetylase-dead mutant YcgC S200A, CobB and the catalytically inactive mutant CobB H110Y in *E. coli* BL21 (DE3) cells and purified them to a high level of purity using a two-step purification strategy composed of a glutathione (GSH)-affinity purification step followed by size exclusion-chromatography (SEC). For YcgC and YcgC S200A Tobacco etch virus (TEV) cleavage was performed to remove the glutathione-S-transferase (GST)-tag. In contrast, CobB and CobB H110Y were purified as GST-fusion proteins to allow for a better discrimination of RutR and CobB in subsequent SDS-PAGE analyses. All proteins could be successfully produced and showed final purities of more than 95% (*Figure 1A*). YcgC and the mutant YcgC S200A behaved similar on analytical SEC further underlining the high quality of proteins (*Figure 1B*, upper panel). Tu et al. reported that YcgC deacetylates the transcriptional regulator RutR acetylated at K52 (RutR AcK52) and at K62 (RutR AcK62). For expression of acetylated RutR, we constructed an *E. coli* BL21 (DE3) strain carrying a genomic deletion of *ycgC* and *cobB* (*E. coli* BL21 (DE3) *ΔycgCΔcobB*) to exclude the possibility that RutR is deacetylated endogenously during the expression (*Figure 1—figure supplement 1*).

We used the genetic code expansion concept to site-specifically introduce acetyl-L-lysine at positions K52 and K62 into RutR. To this end, we constitutively expressed the synthetically evolved acetyl-lysyl-tRNA synthetase *Mb*tRNA$_{CUA}$ pair from *Methanosarcina barkeri* and co-expressed it with RutR carrying an amber stop codon at the positions encoding for K52 and K65 to allow the site-specific incorporation of acetyl-L-lysine. To exclude the possibility that the His$_6$-tag used for purification via $Ni^{2+}$-NTA chromatography interferes with YcgC catalysed deacetylation and subsequent N-terminal autoproteolytic cleavage of RutR, we purified RutR proteins placing the His$_6$-tag at the RutR C-terminus. We were able to produce non-acetylated RutR wildtype (WT), RutR AcK52 and RutR AcK62 purified to high degree and obtained a yield sufficient to perform biochemical studies. Immunoblotting, electrospray ionisation mass spectrometry (ESI-MS) as well as tryptic digest and subsequent ultra-performance liquid chromatography tandem mass spectrometry (UPLC-MS/MS) confirmed that RutR is quantitatively, homogenously and site-specifically lysine acetylated at the positions K52 and K62 (*Figure 1C,D*, *Figure 1—figure supplement 1*). Notably, RutR WT was not acetylated on Lys upon expression in either *E. coli* BL21 (DE3) or in *E. coli* BL21 (DE3) *ΔycgCΔcobB*. This is in contrast to the report by Tu et al., according to which RutR expressed in *E. coli* W3110 is almost quantitatively lysine acetylated (*Figure 1C,D*, *Figure 1—figure supplement 1*). As shown later, our LC-MS/MS data revealed that also the RutR protein from Tu et al. is not quantitatively lysine acetylated. In analytical SEC runs, all RutR proteins eluted not in the exclusion volume, showed a highly symmetric elution peak and eluted at an almost identical elution volume. This shows that all proteins are in a similar oligomeric state and it furthermore suggests that lysine acetylation of RutR does not interfere with protein folding or the oligomeric state (*Figure 1B*, lower panel).

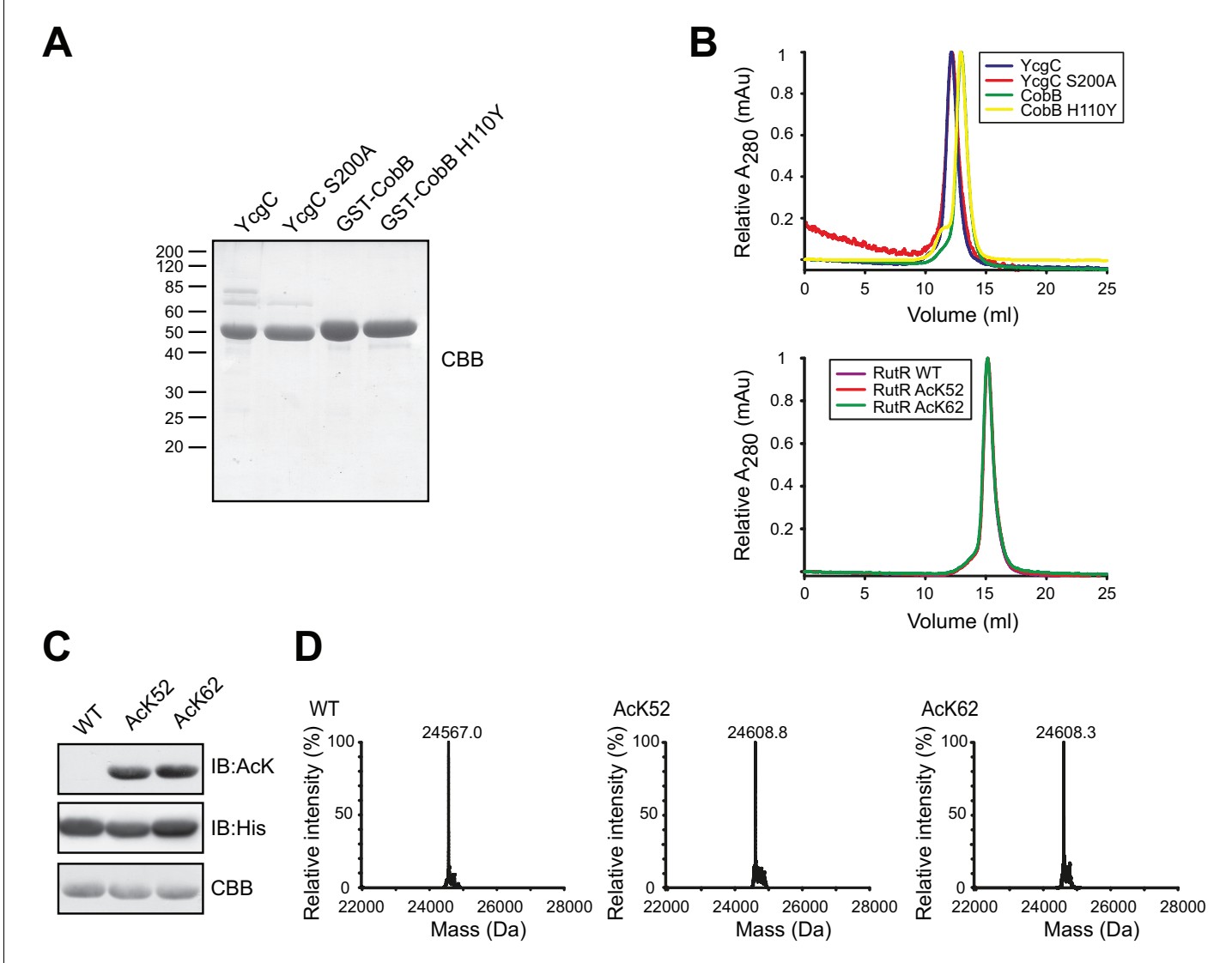

**Figure 1.** Preparation of YcgC, CobB and non-acetylated/acetylated RutR. (A) SDS-PAGE analysis of proteins used in this study. All proteins were expressed and purified as GST-fusion proteins. In terms of YcgC and YcgC S200A the GST-tag was removed by TEV protease during the purification steps. Staining of the gel was done by coomassie brilliant blue (CBB). For molecular masses see figure legend for *Figure 2A*. (B) Analytical size exclusion chromatography on a S200 10/300 GL column shows that YcgC WT and YcgC S200A as well as CobB and the corresponding catalytically inactive variant CobB H110Y display an almost identical elution profile. Moreover, RutR proteins show a nearly identical elution profile in analytical SEC runs indicating that RutR acetylation at K52 and K62 does not interfere with protein folding or its oligomeric state. (C) RutR-His$_6$ AcK52 and AcK62 are quantitatively acetylated. Shown are SDS-PAGE and immunoblot analyses of all RutR-His$_6$ proteins used in this study. Staining for AcK using an anti-acetyl-L-lysine antibody revealed a strong signal for RutR AcK52 and AcK62, whereas no signal was obtained for RutR WT. As loading control anti-His$_6$ staining was performed. (D) ESI-MS data show the quantitative and homogenous incorporation of acetyl-L-lysine into RutR. Shown is the deconvoluted spectrum on the true mass scale after software transformation yielding one single peak and the corresponding molecular mass as indicated. Expected mass non-acetylated RutR: 24567.5 Da; acetylated RutR: 24609.5 Da.

DOI: https://doi.org/10.7554/eLife.37798.002

The following figure supplement is available for figure 1:

**Figure supplement 1.** Construction of an *E.coli* BL21 (DE3) Δ*ycgC*Δ*cobB* double knockout strain and tryptic digest and subsequent LC-MS analysis of RutR AcK52 and RutR AcK62.

DOI: https://doi.org/10.7554/eLife.37798.003

## YcgC does not deacetylate RutR AcK52 and AcK62

To check for deacetylase activity of YcgC we used the site-specifically lysine acetylated RutR AcK52 and AcK62 proteins as substrates and performed an in vitro deacetylation assay. YcgC was used in a twofold molar excess to RutR substrate to ensure that the reaction proceeds to completion. As a readout we conducted immunoblotting using a specific anti-acetyl-L-lysine antibody (anti-AcK AB) that we have shown earlier to be well suited for following deacetylation reactions. We did not observe any YcgC catalysed deacetylation of RutR AcK52 or RutR AcK62 (*Figure 2A*). As a control, we also analysed the deacetylation of RutR AcK52 and AcK62 by CobB. As shown by immunoblotting, RutR AcK52 was strongly deacetylated by CobB whereas RutR AcK62 was only slightly deacetylated (*Figure 2A*). To confirm the immunoblotting results and to exclude the possibility that this assay is not sensitive enough to detect a low deacetylase activity of YcgC on acetylated RutR, we also analysed the reaction products by ESI-MS. Again, no YcgC deacetylase activity towards either RutR AcK52 or RutR AcK62 was measurable and only one single peak with a mass of 24608 Da exactly corresponding to mono-lysine acetylated RutR was detectable (*Figure 2B*, *Figure 2—figure supplement 1*). In contrast, RutR AcK52 was deacetylated by CobB to more than 60% resulting in a

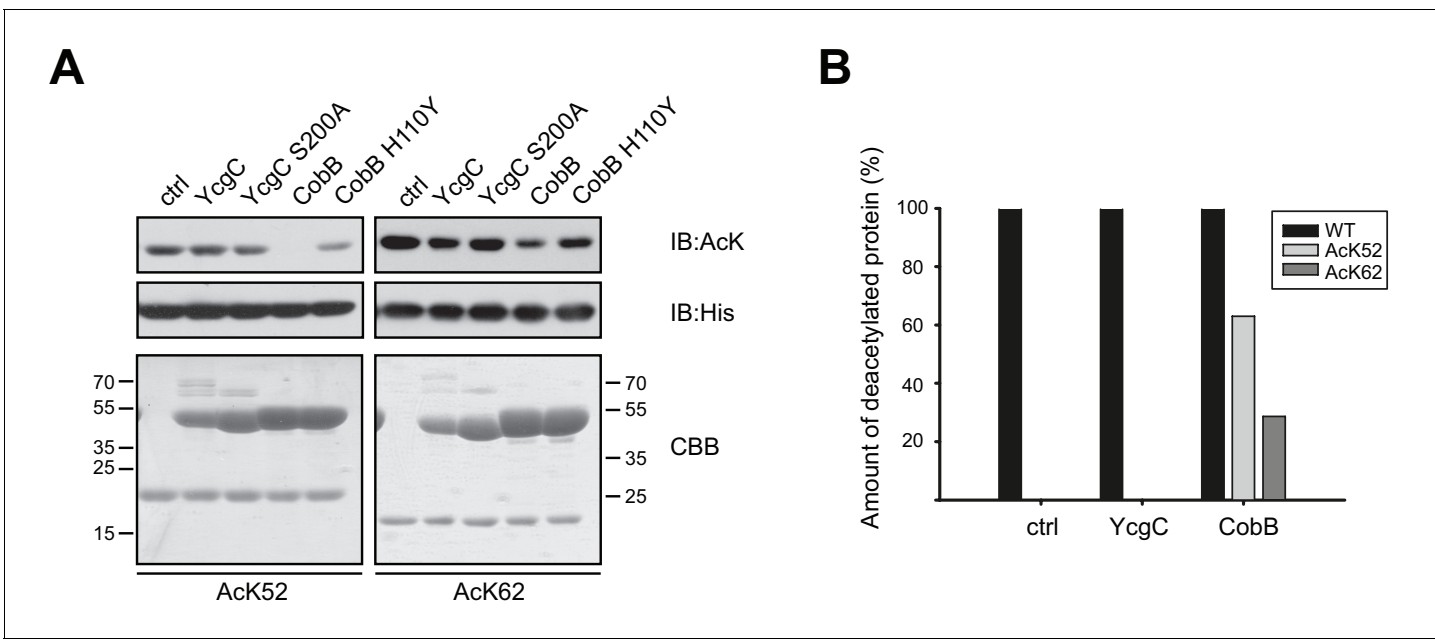

**Figure 2.** YcgC does not show any deacetylase activity towards RutR AcK52 or RutR AcK62 and it does not stimulate RutR (auto-) proteolytic cleavage. (A) RutR AcK52 and RutR AcK62 were treated with a two-fold molar excess of YcgC, YcgC S200A, CobB or CobB H110Y in the presence of 5 mM NAD$^+$. As visible from the immunoblotting using an anti-AcK AB CobB was able to completely deacetylate RutR AcK52 while RutR AcK62 is only faintly deacetylated under the conditions used. YcgC, YcgC S200A and the catalytically dead CobB H110Y did neither deacetylate RutR AcK52 nor RutR AcK62 suggesting that YcgC is no deacetylase for RutR. We used the GST-fusion proteins of CobB and CobB H110Y here to get a better separation of CobB and RutR proteins. Probing of the His$_6$-tag using an anti-His$_6$ antibody was done to show equal loading with RutR proteins. SDS-PAGE analysis and CBB staining show that there is only one band of the same size for RutR for all the conditions tested suggesting that YcgC did also not stimulate (auto-) proteolytic cleavage of RutR. The control lane (ctrl) shows the respective RutR protein, AcK52 or AcK62, without addition of CobB or YcgC. Molecular weights of proteins used (all masses calculated without N-terminal methionine): RutR, 24567.5 Da; acetylated RutR, 24609.5 Da; GST-CobB, 53271.09 Da; GST-CobB H110Y, 53297.12 Da.; YcgC, 51649.80 Da; YcgC S200A, 51633.80 Da. We used the GST-fusion proteins for CobB to obtain a better separation from RutR as CobB (MW: 26314.86 Da) and CobB H110Y (26340.90 Da) without GST-tag have similar molecular weights compared to RutR. (B) Quantification of ESI-MS spectra of YcgC and CobB reaction products shown as immunoblots in (A). As a support for the data obtained by immunoblotting, neither YcgC nor YcgC S200A nor catalytically inactive CobB H110Y did alter the molecular mass of RutR WT, AcK52 and AcK62. However, active CobB led to more than 60% deacetylation of RutR AcK52 while RutR AcK62 was only marginally deacetylated (app. 30% deacetylated). Again, these data confirm that YcgC does neither deacetylate or proteolytically cleave RutR nor does it stimulate autoproteolytic cleavage of RutR.

DOI: https://doi.org/10.7554/eLife.37798.004

The following figure supplement is available for figure 2:

**Figure supplement 1.** Identification and quantification of acetylated and deacetylated RutR by ESI-MS.
DOI: https://doi.org/10.7554/eLife.37798.005

mass shift of 42 Da, which agrees with removal of an acetyl-group, while RutR AcK62 was only marginally deacetylated by CobB (*Figure 2B*, *Figure 2—figure supplement 1*). These data support the immunoblotting experiments and clearly show that YcgC does not have any deacetylase activity towards RutR AcK52 and AcK62. Moreover, we did not observe any (auto-)proteolytic cleavage of RutR WT, RutR AcK52 or RutR AcK62, independently of presence or absence of YcgC. Except for the deacetylation of RutR AcK52, and to a much lesser extent of RutR AcK62, by CobB the molecular weight of the RutR proteins is not affected as shown by SDS-PAGE, immunoblotting and ESI-MS (*Figure 2*, *Figure 2—figure supplement 1*).

## Analysis of YcgC deacetylation of RutR expressed in *E. coli* W3110 (by Tu et al.) and site-specifically lysine-acetylated RutR AcK52 and AcK62 expressed in BL21 (DE3) (this work)

To resolve discrepancies observed between data obtained by Tu and co-workers and by our group we initiated experiments to compare activities of the proteins of both groups (RutR and YcgC; *Figure 3*, *Figure 3—figure supplement 1*, *Supplementary file 5*). To this end, we exchanged all proteins between both laboratories. We performed YcgC catalysed deacetylation reactions under conditions described by Tu et al. (reaction buffer: 50 mM Tris, 4 mM MgCl$_2$, 50 mM NaCl, 50 mM KCl, 5% (v/v) Glycerol, pH 8.0). We also compared the antibodies used for detection in both studies (our lab: ab21263 (abcam); Tu et al.: CST9441 (Cell Signaling Technology)). Using the antibody from abcam we confirmed the results obtained before. We only detected site-specifically lysine-acetylated RutR AcK52 and AcK62, while none of the other proteins were stained with the anti-acetyl-lysine antibody from abcam (ab21263) (*Figure 3A*). Notably, the antibody CST9441 used by Tu et al. detected all RutR proteins used in this study, i.e. non-acetylated RutR prepared by our group, RutR AcK52 and AcK62, as well as RutR from Tu et al. Moreover, while we observed a CobB catalysed decrease in acetylation-level using the antibody from abcam for RutR Ack52 (and more weakly also for AcK62) as expected, the antibody CST9441 did not detect a decrease in acetylation signal intensity upon CobB treatment (*Figure 3B*). This suggests that it detects trace amounts of acetylation occurring at very low stoichiometry most likely non-enzymatically during expression in *E. coli*. These trace amounts must be present even on non-acetylated RutR protein prepared by us. Alternatively, the CST9441 antibody shows cross-reactivity with other antigenic components, for example the RutR protein sequence. In both cases, the antibody CST9441 might not be suitable for these assays. To underline these conclusions, we observed for our RutR proteins no additional lysine acetylations by ESI-MS showing that if further acetylations happen on the RutR protein, these are present at very low, highly sub-stoichiometric levels (*Figure 1D*).

We observe a reduction of the molecular size upon treatment with the YcgC prepared by Tu et al., but neither for YcgC prepared by us, nor by YcgC S200A prepared by Tu et al., suggesting that the YcgC preparation from Tu et al. contains an activity explaining this observation (*Figure 3A*, *Figure 3—figure supplement 1*). This is true for RutR from Tu et al. as well as for all RutR proteins prepared by us, that is non-acetylated RutR and RutR AcK52 and AcK62. This proteolytic modification must occur from the N-terminus as our C-terminally His$_6$-tagged protein still gives a signal in His$_6$ staining, while N-terminally His$_6$-tagged RutR from Tu et al. completely looses His$_6$-staining. Importantly, for both RutR AcK52 and AcK62 the lower molecular weight RutR protein is still lysine acetylated as shown by immunoblotting using the anti-acetyl lysine antibody from abcam suggesting that both proteins are not deacetylated on either AcK52 or AcK62 prior to proteolytic cleavage. This clearly shows that the autoproteolytic cleavage of RutR is not preceded by the deacetylation of RutR AcK52 and/or AcK62, as was suggested by Tu et al. As additional support, we are able to express and purify non-acetylated RutR in full-length without observing truncation at the N-terminus.

## Mass-spectrometry of YcgC and RutR proteins

As the results above clearly show that YcgC does not have an intrinsic lysine deacetylase activity, it is still an unresolved question what leads to the reduction of the molecular size of RutR proteins upon incubation with YcgC prepared by Tu et al. We concluded that a unspecific or a RutR specific protease activity might be present in the YcgC preparation of Tu et al., which could explain the observed reduction in molecular size upon treatment with YcgC from Tu et al. To analyse protein preparations for presence of proteases, we performed LC-MS/MS analyses (*Supplementary file 1*). We observed,

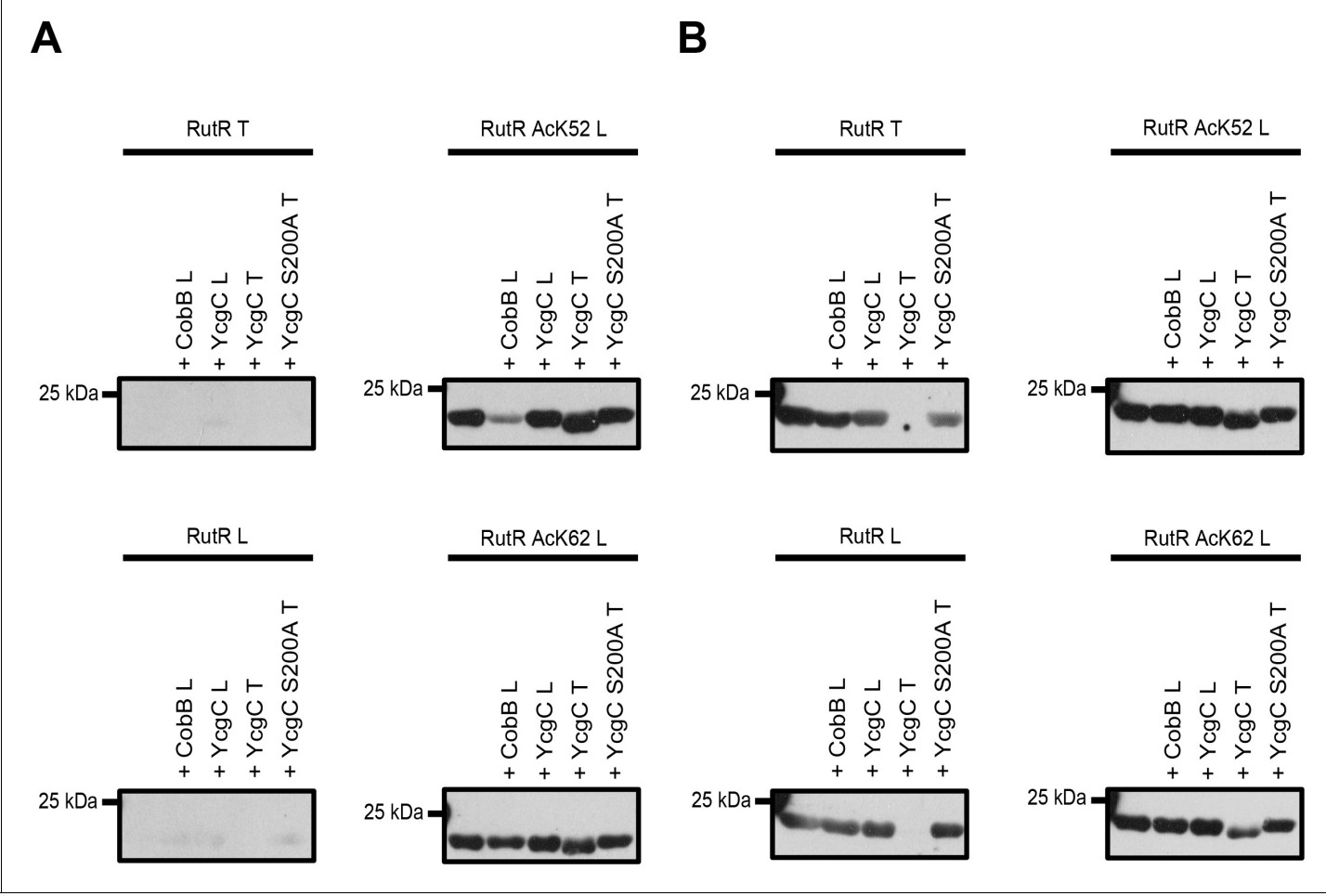

**Figure 3.** Comparison of YcgC and RutR proteins as well as antibody detection from Tu et al. (**T**) and from our lab (**L**). (**A**) RutR from Tu et al. (RutR T) and our non-acetylated RutR (RutR L) were treated with YcgC prepared by our lab (**L**), YcgC and YcgC S200A from Tu et al. (**T**). RutR T and non-acetylated RutR L are not detected by the antibody ab21623 from abcam (left panels). Site-specifically lysine-acetylated RutR AcK52 and AcK62 prepared by us (RutR AcK52 L and RutR AcK62 L) are detected by the antibody ab21623. Treatment with CobB reduces acetylation level for RutR AcK52 L and AcK62 L as expected (right panels). Treatment of RutR AcK52 L and RutR AcK62 L with YcgC T, but not with YcgC S200A T and YcgC L, results in reduction of molecular size, while acetylation level is not decreased. (**B**) Same as in A but staining was done with anti acetyl-lysine antibody CST9441 from Cell Signaling Technology. CST9441 antibody detects site-specifically acetylated and non-acetylated RutR (or RutR that contains highly sub-stoichiometric, trace amounts of lysine acetylation). Acetylation signal is completely removed for RutR T upon treatment with YcgC T but neither with YcgC S200A nor with YcgC L. His$_6$-staining shows that YcgC T treatment also removes N-terminal His$_6$-tag suggesting proteolytic cleavage from N-terminus (*Figure 3—figure supplement 1B*). RutR AcK52 L and AcK62 L also show reduction in molecular size upon treatment with YcgC T (right panels). However, acetylation signal using CST9441 antibody of the degradation band is not removed, showing that deacetylation at neither RutR AcK52 L nor RutR AcK62 L takes place. CobB treatment of no RutR protein results in visible reduction in acetylation signal using CST9441 antibody, although it is shown that CobB deacetylates AcK52 and AcK62 from RutR.

DOI: https://doi.org/10.7554/eLife.37798.006

The following figure supplement is available for figure 3:

**Figure supplement 1.** Loading controls of the deacetylation assays performed under conditions described by Tu et al.

DOI: https://doi.org/10.7554/eLife.37798.007

that the YcgC preparation by Tu et al. contained several contaminants that were neither present in the YcgC S200A preparation by Tu et al. nor in our YcgC protein preparation (*Supplementary file 1*, *Supplementary file 2*). Amongst these proteins, we found several proteases and peptidases (metallo- and serine proteases) and several proteins with unknown function (*Supplementary file 1*, *Supplementary file 2*). Notably, we even detected Lon protease in YcgC by Tu et al., which is genomically deleted in most commercially available bacterial expression strains to avoid proteolytic

cleavage of expressed proteins. One of Lon protease's best-characterised substrates is the helix-turn-helix (HTH) LuxR-type transcriptional regulator RscA. This fuels the idea that maybe the HTH TetR-type transcriptional repressor RutR might be a specific substrate as well. This needs further investigation.

We also used LC-MS/MS to analyse the RutR preparation of Tu et al., and can draw several conclusions from the results of this analysis. First, the RutR protein is not quantitatively lysine acetylated, as was reported by Tu et al. In fact, from our mass spectrometrical analyses we can conclude that overall the amount of acetylated RutR is 1.09% (total intensity of $5.74*10^{11}$ vs. intensity of acetylated peptides: $6.26*10^9$). These analyses can be used to estimate lysine acetylation occupancy. We obtained an overall sequence coverage of 90% in the LC-MS/MS experiments showing that almost all lysines present in RutR are detectable (*Supplementary file 3*). Moreover, although we cannot determine precise RutR-acetylation stoichiometries by LC-MS/MS without performing a standard curve, these data clearly show that the RutR acetylation is of very low occupancy. Acetylated and non-acetylated peptides is reported to show similar ionization efficiencies and as a consequence the MS intensities can be used for quantification of the occupancy (*Cho et al., 2016*). From the identified acetylated lysines, RutR K150 is the acetyl acceptor site with highest occupancy of about 0.7% of total RutR. Lysine 21 is the second best acetyl acceptor site with an occupancy of 0.3% followed by AcK95 with 0.06%. Second, we found AcK52 in the RutR preparation from Tu et al. but again in a very low occupancy of only 0.02% in relation to all peptides identified. Third, while Tu and co-workers identified K62 in RutR as the acetylation site of highest functional importance, our LC-MS/MS data revealed that K62 is not lysine-acetylated at all, although this lysine has been identified in the non-acetylated state and in a peptide with one missed cleavage suggesting that the peptide would be analysable by LC-MS/MS if present. These data clearly show that the model presented by Tu et al. needs to be revised.

## Discussion

In summary, we have shown that YcgC does not show any deacetylase activity towards site-specifically lysine acetylated RutR. Additionally, YcgC prepared in our lab did neither directly nor indirectly affect RutR proteolytic cleavage or autoproteolysis. LC-MS/MS analyses revealed that RutR from Tu et al. is only marginally lysine-acetylated with an overall occupancy of 1%. Of the identified lysine acetylation sites, AcK52 has an occupancy of only 0.02% and AcK62 was not identified at all. For our site-specifically lysine-acetylated RutR proteins, we also observe proteolytic cleavage by treatment with the YcgC preparation from Tu and co-workers. However, also the lower molecular weight RutR proteins were lysine-acetylated.

Taken together, these data call into question the molecular model presented by Tu et al.: in other words, the data show that the (auto-)proteolysis of RutR is not preceded by the YcgC-catalysed deacetylation of RutR at AcK52 and/or AcK62. Our LC-MS/MS analyses revealed the presence of a number of contaminants, including some that have proteolytic activity, in the YcgC preparation provided by Tu et al. Finally, threading analyses of YcgC using the programme iTASSER revealed no obvious homologies to enzymes with deacetylase or protease activity (*Supplementary file 4*) (*Yang et al., 2015*).

## Materials and methods

### *Generation of E. coli BL21 (DE3)ΔycgCΔcobB*

The double knockout strain *E. coli BL21 (DE3) ΔycgCΔcobB* was created by applying the Quick and Easy *E. coli* Gene Deletion Kit (Gene Bridges). Consecutive deletion of *ycgC* and *cobB* was done by homologous recombination according to the manufacturer's constructions. Finally, successful gene deletion was verified by PCR and sequencing.

### Expression and purification of proteins

*E. coli* YcgC, YcgC S200A, CobB and CobB H110Y were expressed as GST-fusion proteins using the vector pGEX-4T5/TEV derived from pGEX-4T1 (GE Healthcare). RutR proteins were expressed as fusion proteins carrying a C-terminal $His_6$-tag using the vector pRSF-Duet-1. YcgC, YcgC S200A,

CobB and CobB H110Y were expressed in *E. coli* BL21 (DE3), whereas RutR proteins were expressed in *E. coli* BL21 (DE3) $\Delta ycgC\Delta cobB$. Cells were grown to an optical density at 600 nm (OD$_{600}$) of 0.6 (37°C, 160 rpm) before protein expression was induced by addition of 300 µM isopropyl-β-D-thioga-lactopyranoside (IPTG). Protein expression was performed over night at 18–20°C at 160 rpm. Subsequently, the bacterial cultures were harvested (4000 g, 10 min) and resuspended in either (YcgC and CobB) buffer A (50 mM Tris/HCl pH 7.4, 100 mM NaCl, 5 mM MgCl$_2$, 2 mM β-mercaptoethanol) or (RutR) buffer B (100 mM K$_2$HPO$_4$/KH$_2$PO$_4$ pH 6.4, 150 mM NaCl, 20 mM imidazole, 2 mM β-mercap-toethanol) containing 200 µM Pefabloc protease inhibitor cocktail. Cells were lysed by sonication and after centrifugation (20000 g, 45 min) the soluble fraction was applied to an equilibrated GSH- or Ni$^{2+}$-NTA affinity chromatography column. Washing was performed with either buffer A (YcgC and CobB) or buffer B (RutR) supplemented with 500 mM NaCl. YcgC GST-fusion proteins were treated with TEV protease on the GSH-column to remove the GST-tag (4°C, 0.5 ml/min, over night). CobB was either eluted as GST-fusion protein with buffer A containing 30 mM glutathione or it was treated with TEV protease to remove the GST-tag. RutR proteins were eluted as GST-fusion proteins with buffer C (buffer B with imidazole gradient from 50 to 300 mM imidazole). Eluates were concentrated using 10 kDa MWCO amicon ultrafiltration devices and subsequently applied to a size exclusion chromatography (SEC) column (GE healthcare). SEC was performed in buffer A (YcgC and CobB) or buffer B (RutR), respectively. Peak fractions containing the protein of interest were pooled, shock frozen in liquid nitrogen and stored at −80°C. Protein concentrations were determined by measuring the absorption at 280 nm using the protein's extinction coefficient (http://web.expasy.org/protparam/).

## Incorporation of N-(ε)-acetyl-lysine into RutR at K52 and K62 using the genetic code expansion concept

The site-specific incorporation of N-(ε)-acetyl-lysine into RutR at position K52 and K62 was conducted as described previously (*de Boor et al., 2015*; *Knyphausen et al., 2016*; *Kuhlmann et al., 2016*; *Lammers et al., 2010*; *Neumann et al., 2008*). In brief, *E. coli* BL21 (DE3) $\Delta ycgC\Delta cobB$ cells bearing the vector pRSF-Duet-1 encoding for RutR K52amber or RutR K62amber, the synthetically evolved acetyl-lysyl-tRNA-synthetase AcKRS3 and the amber suppressor tRNA$_{CUA}$, *Mb*tRNA$_{CUA}$ derived from *Methanosarcina barkeri* were cultivated to an OD$_{600}$ of 0.6 (37°C, 160 rpm). After addition of 10 mM N-(ε)-acetyl-L-lysine (Chem-Impex International Inc.) and 20 mM nicotinamide (NAM) cells were grown for further 30 min before protein expression was induced by adding 300 µM IPTG. The quantitative incorporation of N-(ε)-acetyl-L-lysine into RutR at position K52 and K62 was verified by immunoblotting using an anti-acetyl-L-lysine antibody and mass spectrometry as described earlier (*de Boor et al., 2015*; *Lammers et al., 2010*).

### Immunoblotting

Proteins were separated by SDS-PAGE and analysed by immunoblotting using a standard protocol. Bound antibodies were visualised by using enhanced chemiluminescence (Roth). Rabbit polyclonal anti-acetyl-L-lysine antibody (ab21623, 1:1500), mouse monoclonal anti-His$_6$ antibody (ab18184, 1:2000), rabbit monoclonal anti-acetyl-L-lysine antibody (CST9441, 1:1000) as well as HRP-coupled secondary antibodies against rabbit (ab6721, 1:10000) and mouse (ab6728, 1:10000) were purchased from Abcam.

### Deacetylase assay

To test for deacetylation of RutR-His$_6$ AcK52 and RutR-His$_6$ AcK62 by YcgC and GST-CobB 12.5 µM of recombinant RutR-His$_6$ were incubated with 25 µM YcgC, YcgC S200A, GST-CobB or GST-CobB H110Y. All reactions were done in buffer A supplemented with 5 mM NAD$^+$. To this end, all proteins were transferred into buffer A by using 10 kDa MWCO Viaspin 500 microcentrifugal units. After incubation for one hour at 37°C, either samples were boiled at 95°C for five minutes and subjected to SDS-PAGE and immunoblotting or reaction products were analysed by ESI-MS.

For comparison of proteins from Tu et al. and our group (Tu et al.: YcgC T, YcgC S200A T, RutR T; Kremer and Kuhlmann et al.: YcgC L, RutR L, RutR AcK52 L, RutR AcK62 L) to resolve discrepancies between results, we performed deacetylation assays in the exact buffer as described in Tu et al. (50 mM Tris, 4 mM MgCl$_2$, 50 mM NaCl, 50 mM KCl, 5% (v/v) Glycerol, pH 8.0). We incubated 6.3

µg (≈ 25 µM) of the respective RutR protein with 6.3 µg (≈ 12 µM) YcgC T, YcgC S200A T, YcgC L or 3.15 µg (12 µM) CobB filled up with buffer (and NAD$^+$ for CobB) to a total volume of 10 µL for 2 hr at 37°C. Analysis of the lysine acetylation status, the molecular size of RutR and protein loading was done by immunoblotting using an anti-acetyl-lysine antibody from abcam (ab21623) or Cell Signaling Technology (CST9441), anti-His$_6$-antibody (ab18184, 1:2000) and Coomassie-brilliant blue (CBB) staining.

## Electrospray-ionisation (ESI)-mass spectrometry

To verify successful incorporation of acetyllysine, intact proteins were analysed by ESI-MS on a LTQ Orbitrap Discovery mass spectrometer (Thermo Scientific). To validate specificity of the acetylation site, RutR was trypsin-digested and desalted prior to analysis by UPLC-MS/MS. In brief, 4 µg protein were denatured in urea buffer (50 mM TEAB, 8 M urea) and reduced by adding dithiothreitol (DTT) to a final concentration of 5 mM and incubation at 37°C for one hour. Subsequently, proteins were alkylated with chloroacetamide (CAA) (40 mM, 30 min in the dark) followed by LysC digestion for 4 hr at 37°C. Afterwards, tryptic digest was done over night at 37°C. For both enzymes an enzyme to substrate ratio of 1:75 was used. The next day, obtained peptides were acidified by adding formic acid to a final concentration of 1%, desalted and analysed with a Q Exactive Plus Orbitrap LC-MS/MS system (Thermo Scientific) as described earlier (*Kuhlmann et al., 2016*). MaxQuant and the implemented Andromeda search engine were used to analyse raw data. MS/MS spectra were correlated with the Uniprot *Escherichia coli* database containing the target protein (*Cox and Mann, 2008*; *Cox et al., 2011*).

## Liquid chromatography (LC)-mass spectrometry and data analysis

Peptides were analysed on a Q Exactive Plus (Thermo Scientific) mass spectrometer that was coupled to an EASY nLC 1000 UPLC (Thermo Scientific). Samples were loaded with solvent A (0.1% formic acid in water) onto an in-house packed analytical column (50 cm × 75 µm I.D., filled with 2.7 µm Poroshell EC120 C18, Agilent). Peptides were chromatographically separated at a constant flow rate of 250 nL/min using 60 min methods: 5–30% solvent B (0.1% formic acid in 80% acetonitrile) within 40 min, 30–50% solvent B within 8 min, followed by washing and column equilibration. The mass spectrometer was operated in data-dependent acquisition mode. The MS1 survey scan was acquired from 300 to 1750 m/z at a resolution of 70,000. The top 10 most abundant peptides were subjected to HCD fragmentation at a normalised collision energy of 27%. The AGC target was set to 5e$^5$ charges. Product ions were detected in the Orbitrap at a resolution of 17,500. All mass spectrometric raw data were processed with Maxquant (version 1.5.3.8) using default parameters [*Tyanova et al., 2016*]). Briefly, MS2 spectra were searched against the *E. coli* UniProt database, including a list of common contaminants and sequences of the specific purified proteins in the samples. False discovery rates on protein and PSM level were estimated by the target-decoy approach to 0.01% (Protein FDR) and 0.01% (PSM FDR) respectively. The minimal peptide length was set to seven amino acids and carbamidomethylation at cysteine residues was considered as a fixed modification. Oxidation (M), Acetyl (K) and Acetyl (Protein N-term) were included as variable modifications. The resulting output was processed using Perseus as follows: Protein groups flagged as 'reverse', 'potential contaminant' or 'only identified by site' were removed from the proteinGroups.txt. Intensity or iBAQ values were log transformed.

## Acknowledgements

We thank the CECAD proteomics facility headed by Dr. Christian Frese and Dr. Stefan Müller for the helpful discussions setting up the MS experiments and conducting the experiments. This work was funded by Cologne Excellence Cluster on Cellular Stress Responses in Aging-Associated Diseases (CECAD), the Heisenberg Programm (grant: LA2984/3-1) of the German Research Foundation (Deutsche Forschungsgemeinschaft; DFG) and the Boehringer Ingelheim Foundation (Exploration Grant Programme).

## Additional information

### Funding

| Funder | Grant reference number | Author |
| --- | --- | --- |
| Deutsche Forschungsgemeinschaft | LA 2984/3-1 | Michael Lammers |
| Deutsche Forschungsgemeinschaft | CECAD | Magdalena Kremer<br>Kuhlmann Nora<br>Marius Lechner<br>Linda Baldus<br>Michael Lammers |
| Boehringer Ingelheim Fonds | Exploration Grant | Kuhlmann Nora |
| Deutsche Forschungsgemeinschaft | Cologne Graduate School of Ageing Research | Magdalena Kremer |

The funders had no role in study design, data collection and interpretation, or the decision to submit the work for publication.

### Author contributions

Magdalena Kremer, Conceptualization, Formal analysis, Methodology; Nora Kuhlmann, Marius Lechner, Investigation, Methodology; Linda Baldus, Investigation; Michael Lammers, Conceptualization, Formal analysis, Supervision, Funding acquisition, Validation, Investigation, Methodology, Writing—original draft, Project administration, Writing—review and editing

### Author ORCIDs

Michael Lammers (iD) http://orcid.org/0000-0003-4168-4640

### Decision letter and Author response

Decision letter https://doi.org/10.7554/eLife.37798.015
Author response https://doi.org/10.7554/eLife.37798.016

## Additional files

### Supplementary files

• Supplementary file 1. The dataset contains all proteins identified in preparations of recombinantly expressed YcgC and YcgC S200A from Tu et al. and of YcgC prepared by us.
DOI: https://doi.org/10.7554/eLife.37798.008

• Supplementary file 2. Proteases identified in the YcgC sample of Tu et al., which are neither present in the catalytically inactive YcgC S200A preparation from Tu et al. nor in the YcgC preparation from Kremer and Kuhlmann et al.
DOI: https://doi.org/10.7554/eLife.37798.009

• Supplementary file 3. The dataset contains both all proteins identified in preparation of recombinantly expressed RutR from Tu et al. as well as acetylated and non-acetylated peptides from RutR from Tu et al. We observed an overall sequence coverage of almost 93%. The only sequences not identified were 8-TTGKRSRAVSAK-19 and 100-LK-101, most likely because trypsin cleavage generates small peptides not detected by LC-MS/MS. As a result, this shows that there is no acetylation occurring at those lysines.
DOI: https://doi.org/10.7554/eLife.37798.010

• Supplementary file 4. The YcgC amino acid sequence was analysed by the threading programme I-iTASSER. Usage of the structural alignment programme TM-align to match the first I-TASSER model to all structures in the PDB. The following table shows the top ten proteins that show the closest structural similarity to the predicted I-TASSER model. Due to the structural similarity these proteins often have similar function as the target.
DOI: https://doi.org/10.7554/eLife.37798.011

• Supplementary file 5. Protein sequences of recombinantly expressed RutR and YcgC proteins from Tu et al.
DOI: https://doi.org/10.7554/eLife.37798.012

**Data availability**

All data generated or analyses during this study are included in the manuscript and supporting files.

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
