## [Decision Letter]

Thank you for submitting your work entitled "Comment on "YcgC represents a new protein deacetylase family in prokaryotes"" for consideration by *eLife*. Your article has been reviewed by three peer reviewers including Wilfred van der Donk as the Reviewing Editor and Reviewer #2, and the evaluation has been overseen by Michael Marletta as the Senior Editor. The reviewers have opted to remain anonymous.

Our decision has been reached after consultation between the reviewers. All three reviewers recognize the importance to publish your work if indeed the previous assignment of YcgC as a deacetylase turns out to be incorrect. However, all three reviewers also believe that at present, your experiments have not used the same conditions and enzymes as those used in the original work. Hence, the discrepancies could well lie in those differences in conditions and enzymes. Based on these discussions and the individual reviews below, we regret to inform you that your work as submitted will not be considered further for publication in *eLife*. However, if you were to investigate the deacetylase activity of YcgC on acetylated RutR under the same conditions as the previous study and find that no activity is observed, we would be willing to reconsider. The journal would also like to help in exchange of materials with the initial authors to find the source of the discrepancies.

Reviewer #1:

In this manuscript, Kuhlmann et al. reported the function characterization of YcgC, which conflicts with what was reported by Tu and co-workers before. The negative results of the authors led to their claim that "YcgC is not a protein deacetylase". After comparing the two manuscripts, the reviewer found out the following key differences. On the basis of the limited data shown here, the reviewer could not decide whether the listed difference led to the conflicting results presented in the two manuscripts. However, given that the current manuscript built up its argument on the basis of the negative results and reported the finding after the first report, it should be the burden of the current authors to rule out that any of the following differences or their combination is not the cause of the conflicting observation. If possible, the authors should contact the lab for identical reagents unless it is inaccessible. It is not sufficient to simply justify why the new conditions were used without conducting the same experiments under the identical conditions as reported previously.

a) Buffer conditions for deacetylase assay:

Current manuscript: 50 mM Tris-HCl pH 7.4, 100 mM NaCl, 5 mM MgCl_2_, 5 mM NAD^+^, 2 mM β-mercaptoethanol, 12.5 μm substrate and 25 μm YcgC at 37°C for 1 hr. The manuscript of Tu: 50 mM Tris-HCl pH 8.0, 50 mM NaCl, 4 mM MgCl_2_, 50 mM KCl, 1 mM NAD^+^, 0.15 μg/μl substrate, and 2.25 μg/μl enzyme at 37 °C for 1 hr.

b) RutR Substrates:

Current manuscript: C-terminal His6-tag containing either K52Ac or K62Ac expressed in *E. coli* BL21 (DE3) *ΔycgCΔcobB* strain through amber-suppressor-mediated non-natural amino acid incorporation.

The manuscript of Tu: C-terminal 3xFLAG-tagged expressed in *E. coli* strain (W3110) with K52Ac and K62Ac through posttranslational modifications.

c) YcgC:

Current manuscript: GST-TEV-tagged-YcgC in combination with TEV cleavage.

The manuscript of Tu: GST-YcgC

d) Antibody readouts:

Current manuscript: a specific anti-acetyl-L-lysine antibody (anti-AcK AB) that we have shown earlier to be well suited for following deacetylation reactions (de Boor, Knyphausen et al., 2015, Knyphausen et al., 2016, Kuhlmann, Wroblowski et al., 2016). The manuscript of Tu: an α-AcK antibody (Cell Signaling Technology, Shanghai, China)

Reviewer #2:

In this study, Lammers and coworkers report experiments that potentially call into question the conclusions of a previous study reported in *eLife* by Tu et al. In that previous paper, the authors reported that YcgC is a new type of deacetylase and that YcgC regulates transcription by catalyzing deacetylation of Lys52 and Lys62 of a transcriptional repressor RutR. The work was exciting because YcgC is not related to sirtuins or classical Lys deacetylases, and would add a new type of bacterial deacetylase (*E. coli* has only a single validated deacetylase, CobB). The proposed YcgC activity was discovered by using a clip-chip strategy, and activity was supported with His-tagged YcgC expressed in *E. coli* and purified and acetylated RutR obtained from an *E. coli* strain that apparently results in strongly acetylated protein. I read the previous paper and its initial reviewer comments and the response and I must say that the previous study appears solid.

In the current work, YcgC was expressed as a GST-tagged protein in *E. coli*, purified, and the tag removed by TEV protease, and C-terminally His-tagged RutR was obtained acetylated at the previously reported Lys52 and Lys62 by stop-codon suppression methods and purified. Unfortunately, in vitro incubation of the putative deacetylase and acetylated substrate did not lead to any activity.

If indeed YcgC is not a deacetylase, it will be very important for the community and hence I believe this work is important. However, there could be explanations for the discrepancies other than that the previous study was flawed, and these need to be explored. Both the YcgC enzyme and its substrate appeared to have been prepared differently in this study than in the previous study and apparently with different affinity tags. To truly test whether the previous work is reproducible, the same constructs should be used. The observations in the current study that the protein "behaved similarly on analytical SEC" is not really a measure of "the high quality of proteins". It shows that the proteins behave the same, but says nothing about the "quality". Similarly, I don't think that attaching the His-tag at the C-terminus of YcgC ensures it retains its activity as is written in the second paragraph of the subsection “Expression and purification of YcgC, CobB and RutR”, and identical behavior on SEC says nothing about proper folding, just that they are folded the same. I do not think the authors can rule out that either YcgC or RutR may be improperly folded in the current study, or that the hyper acetylation of RutR in the previous work may have been important.

I think at minimum, the authors of the current paper should prepare the same form of YcgC and the same form of RutR as in the previous study (same affinity tags and at the same positions). *eLife* should assist in requesting materials from Tu et al. In the end it is in the interest of both research groups and of the research community that the conclusions of the previous study are tested, but it is not in the interest of anyone if it turns out that the differences in conditions/proteins caused the apparent discrepancy and that YcgC does in fact have deacetylase activity.

When I read the previous study by Tu et al. I could not find if their proteins for their in vitro data were His-tagged at the N- or C-terminus. This is critical information and if it indeed was not in the paper, *eLife* should request that information from the original authors.

I did not think that the section: "YcgC does not show any deacetylase or proteolytic activity" added much if any value. Tu et al. never suggested YcgC was a general protease like trypsin. Instead they clearly proposed that upon deacetylation (by either YcgC or by CobB) RutR undergoes autoproteolysis. They also never suggested YcgC to act on a p53 based peptide. Hence, I did not think that these studies had much merit. The focus should be on YcgC-catalyzed deacetylation of RutR with proteins that are the same as those in the Tu et al. study. If that indeed leads to no activity, then I think it is important to publish this study.

Reviewer #3:

This report by Lammers et al. rebutted the previous discovery by Tu et al. that Ycgc is a unique protein deacetylase in bacteria.

Subsection “Expression and purification of YcgC, CobB and RutR”, first paragraph: GSH should be GST.

Figure 2A: in this reviewer's visual judgement, YcgC had some deacetylase activity but YcgC S200A did not.

Figure 2B was poorly presented and easy to cause confusion. The authors should include data of RutR WT, AcK52 and AcK62 for each cluster (i.e. Control, YcgC, and CobB). Error bars should be included. The figure legend does not seem to match the figure graph.

YcgC is a bacterial protein but p53 is a mammalian protein; therefore it is not a surprise that p53 peptide is not a substrate of YcgC. This point is particularly important as the previous study pointed out that YcgC has very narrow substrate specificity.

Tu et al. showed that RutR expressed in *E. coli* W3110 was highly acetylated, but this is not the case in this present study where BL21 (DE3) was used. Such drastic difference means physiologies of the two cellular systems are distinct and conclusions on RutR acetylation cannot be compared. This deserves further investigation.

This reviewer is worried about whether the differences in the protein constructs, acetyl-lysine antibody, and experimental assay protocols used between Tu's study and this one, may be the main reasons accounting for the divergent conclusions on whether YcgC has deacetylase activity. YcgC was assumed to have different mechanism from CobB, but why did the authors used the same protocols for deacetylase activity test with NAD present?

[Editors’ note: what now follows is the decision letter after the authors submitted for further consideration.]

Thank you for resubmitting your work entitled "Comment on "YcgC represents a new protein deacetylase family in prokaryotes"" for further consideration at *eLife*. Your revised article has been reviewed by three peer reviewers, one of whom is a member of our Board of Reviewing Editors, and the evaluation has been overseen by Michael Marletta as the Senior Editor.

In this revision, you have carried out experiments to further test whether YcgC is not a deacetylase. You provide data suggesting that the observed activity in the previous *eLife* paper might arise from the trace contamination of protease(s). The evidence presented in the manuscript using both reagents, enzyme, and substrate preparations from the original authors and the current authors is strong. The methods used are rigorous using both MS and western blotting, You also show data suggesting that the original findings were highly dependent on blots using an antibody that appears non-selective for the desired antigen.

The manuscript has been improved and all three reviewers are strongly supportive of publication in *eLife* but you will need to address some editorial points (the article is much longer than our guidelines recommend, and the tone also needs attention in a number of places).

Revisions needed:

In the third paragraph of the Discussion, the authors provide quantification of the amount of acetylation of RutR in the preparation of Tu et al. They use the fact that they achieve 90% sequence coverage as argument that the mass spectrometry intensities can be used for quantification. The reviewers do not think this is valid. If the acetylated peptides ionize much more poorly than the nonacetylated peptides, then peak intensities can only be used if the authors have standard curves of mixtures of different ratios. Hence, they should be more qualitative in their discussions.

In Figure 2A, what is in the ctrl lane? Also, do YcgC and CobB happen to have the same mass? If so, it may be good to provide the calculated molecular weights of RutR, YcgG and CobB in the legend to avoid confusion for readers.

---

## [Author Response]

Our decision has been reached after consultation between the reviewers. All three reviewers recognize the importance to publish your work if indeed the previous assignment of YcgC as a deacetylase turns out to be incorrect. However, all three reviewers also believe that at present, your experiments have not used the same conditions and enzymes as those used in the original work. Hence, the discrepancies could well lie in those differences in conditions and enzymes. Based on these discussions and the individual reviews below, we regret to inform you that your work as submitted will not be considered further for publication in eLife. However, if you were to investigate the deacetylase activity of YcgC on acetylated RutR under the same conditions as the previous study and find that no activity is observed, we would be willing to reconsider. The journal would also like to help in exchange of materials with the initial authors to find the source of the discrepancies.

As we were really inspired by the great findings by Tu and co-workers which reported the identification of a novel lysine-deacetylase class in prokaryotes, we initiated studies to elucidate structure and function to characterise the catalytic mechanism employed by YcgC to deacetylate lysine side-chains of substrates such as RutR suggested to result in its autoproteolytic cleavage.

We used a synthetic biological approach to produce quantitatively and site-specifically lysine acetylated RutR K52 (RutR AcK52) and K62 (RutR AcK62). These are the two by Tu and colleagues described acetyl-acceptor lysines in RutR. Moreover, we recombinantly expressed YcgC and purified it to a high level of purity. However, we were quite surprised that our data robustly show that YcgC was not capable to deacetylate neither RutR AcK52 nor RutR AcK62. In contrast, CobB was a potent deacetylase for RutR AcK52 and less efficiently also for RutR AcK62. We confirmed our data by electrospray-ionization mass-spectrometry next to immunoblotting experiments.

Reviewer #1:

In this manuscript, Kuhlmann et al. reported the function characterization of YcgC, which conflicts with what was reported by Tu and co-workers before. The negative results of the authors led to their claim that "YcgC is not a protein deacetylase". After comparing the two manuscripts, the reviewer found out the following key differences. On the basis of the limited data shown here, the reviewer could not decide whether the listed difference led to the conflicting results presented in the two manuscripts. However, given that the current manuscript built up its argument on the basis of the negative results and reported the finding after the first report, it should be the burden of the current authors to rule out that any of the following differences or their combination is not the cause of the conflicting observation. If possible, the authors should contact the lab for identical reagents unless it is inaccessible. It is not sufficient to simply justify why the new conditions were used without conducting the same experiments under the identical conditions as reported previously.a) Buffer conditions for deacetylase assay:Current manuscript: 50 mM Tris-HCl pH 7.4, 100 mM NaCl, 5 mM MgCl_2_, 5 mM NAD^+^, 2 mM β-mercaptoethanol, 12.5 μm substrate and 25 μm YcgC at 37°C for 1 hr. The manuscript of Tu: 50 mM Tris-HCl pH 8.0, 50 mM NaCl, 4 mM MgCl_2_, 50 mM KCl, 1 mM NAD^+^, 0.15 μg/μl substrate, and 2.25 μg/μl enzyme at 37 °C for 1 hr.

We repeated the experiments in exactly the same buffer and used the original proteins from Tu et al., which we got from Tu and co-workers. However, apart from reduction in molecular size, maybe due to proteolytic cleavage, we could not detect any deacetylase activity. We observed cleavage for all RutR preparations from Tu et al. and from our lab, including our site-specifically acetylated RutR AcK52 and AcK62 proteins. However, the acetylation signal was not removed showing that deacetylation and (auto-)proteolytic cleavage are not coupled. Furthermore, treatment with catalytically active CobB did not show any proteolytic degradation of no RutR protein analysed, including RutR from Tu et al. We also performed LC-MS/MS analyses on the original proteins from Tu et al. and observed that RutR is not quantitatively lysine acetylated with an overall acetylation occupancy of app 1%. AcK62 was not identified as being lysine acetylated. In the preparation of YcgC of Tu and co-workers, we identified several contaminants with protease/peptidase activity not in the YcgC S200A preparation from Tu et al., and also not in our YcgC preparation.

b) RutR Substrates:Current manuscript: C-terminal His6-tag containing either K52Ac or K62Ac expressed in E. coli BL21 (DE3) ΔycgCΔcobB strain through amber-suppressor-mediated non-natural amino acid incorporation. The manuscript of Tu: C-terminal 3xFLAG-tagged expressed in E. coli strain (W3110) with K52Ac and K62Ac through posttranslational modifications.

Tu and co-workers used recombinantly expressed and purified RutR protein carrying an His_6_tag (see sequences in supplemental section of the revised manuscript, which we got from Tu eat al.). We observed the same results independently from where the His_6_-tag is placed, either at RutR N-terminus or at C-terminus.

We examined the RutR preparation of Tu et. Al by LC-MS/MS and found, firstly, that their RutR protein is not quantitatively lysine acetylated but total acetylation stoichiometry is approximately 1%. Secondly, AcK21 and AcK150 are the major acetylation sites, although both also occur at very low occupancy with 0.3% and 0.7%. K52 is found to be lysineacetylated but only with an overall stoichiometry of 0.02% and K62 is not acetylated at all. The working model presented by Tu and co-workers cannot be correct if acetylation is not quantitative. Considering these occupancies, almost quantitative proteolytic degradation of RutR as observed by treatment with YcgC from Tu et al. cannot be explained.

We also examined YcgC and YcgC S200A preparations from Tu and co-workers and found that in their YcgC preparation there are many contaminants not present in YcgC S200A or our YcgC preparation. Amongst those we identified several proteases including Lon protease, which is genetically deleted in most bacterial expression strains.

c) YcgC:Current manuscript: GST-TEV-tagged-YcgC in combination with TEV cleavage.The manuscript of Tu: GST-YcgC

According to communication with Tu et al., they used YcgC/YcgC S200A with N-terminal His_6_-tag as shown in supplemental material in the revised manuscript.

d) Antibody readouts:Current manuscript: a specific anti-acetyl-L-lysine antibody (anti-AcK AB) that we have shown earlier to be well suited for following deacetylation reactions (de Boor, Knyphausen et al., 2015, Knyphausen et al., 2016, Kuhlmann, Wroblowski et al., 2016). The manuscript of Tu: an α-AcK antibody (Cell Signaling Technology, Shanghai, China)

Tu et al. used an anti-acetyl-lysine antibody from Cell Signaling Technology (CST9441). We applied anti-acetyl-lysine antibody (ab21623) from abcam, which we used in several studies in our lab. We compared both antibodies for detection of our protein preparations and protein preparations from Tu and co-workers. We found that the abcam antibody ab21623 did, in contrast to the antibody CST9441, not stain non-acetylated RutR proteins, neither from our lab nor the RutR protein from Tu et al.. This supports the notion that RutR from Tu et al. is not quantitatively lysine-acetylated and CST9441 might be unsuitable for this study. Using the abcam antibody ab21623 we see downshifted site-specifically acetylated RutR AcK52 and AcK62 proteins upon treatment with YcgC from Tu et al.. However, the lower molecular weight band is still lysine-acetylated. This shows that deacetylation by YcgC is not the molecular event preceding RutR (auto-)proteolytic cleavage as postulated by Tu et al.

Importantly, only the abcam antibody ab21623 detects deacetylation catalysed by CobB as the signal decreases for RutR AcK52 and to lesser extent also for AcK62 (which is supported by our ESI-MS data). According to the CST9441 antibody, treatment with CobB does not affect the overall signal of our quantitatively lysine-acetylated proteins RutR AcK52 (nor for AcK62). The CST9441 antibody detects all RutR proteins, from our lab and from Tu et al. Our mass-spec data show that RutR from Tu et al. is acetylated but acetylation occurs highly sub-stoichiometrically. Our hypothesis is that the CST9441 either detects trace amounts of acetyl-lysine and is therefore much more sensitive than the abcam antibody (or it detects something else such as the RutR protein sequence or N-terminal acetylation or a combination of all three). Apparently, if the CST9441 antibody detects lysine acetylation, it detects trace amounts of lysine-acetylation on RutR that could occur non-enzymatically during recombinant expression in *E. coli*. However the total occupancy of acetylation is so low, that it is most likely not of any physiological significance and it can for sure not explain the observed almost quantitative reduction in molecular size upon treatment with (only) YcgC from Tu et al. of all RutR proteins used in these studies (prepared by Tu et al. and by our lab). In sum, we can conclude that for these purposes to draw conclusions on lysine-acetylation and deacetylation the CST9441 antibody is not suitable.

From our point of view, the data can be explained by a protease contamination present in the YcgC preparation. We found many possible candidates selectively only in the YcgC preparation from Tu et al. by LC-MS/MS (but not in YcgC S200A from Tu et al. and not in our YcgC preparation). Proteolytic degradation of RutR might take place, which removes parts of the proteins, that are major sub-stoichiometric lysine-acceptor sites, K21 and K150, in RutR so that they are not even detectable by the antibody CST9441. One candidate might be the protease Lon. One of the Lon protease’s best-characterised substrates is the helix-turn-helix (HTH) LuxR-type transcriptional regulator RscA. This fuels the idea that maybe the HTH TetR-type transcriptional repressor RutR might be a specific substrate as well.

YcgC is clearly no deacetylase and the model must be revised. AcK52 is of very low stoichiometry in preparation of RutR from Tu et al., AcK62 is not present at all and total acetylation occupancy is very low.

Reviewer #2:

In this study, Lammers and coworkers report experiments that potentially call into question the conclusions of a previous study reported in eLife by Tu et al. In that previous paper, the authors reported that YcgC is a new type of deacetylase and that YcgC regulates transcription by catalyzing deacetylation of Lys52 and Lys62 of a transcriptional repressor RutR. The work was exciting because YcgC is not related to sirtuins or classical Lys deacetylases, and would add a new type of bacterial deacetylase (E. coli has only a single validated deacetylase, CobB). The proposed YcgC activity was discovered by using a clip-chip strategy, and activity was supported with His-tagged YcgC expressed in E. coli and purified and acetylated RutR obtained from an E. coli strain that apparently results in strongly acetylated protein. I read the previous paper and its initial reviewer comments and the response and I must say that the previous study appears solid.

We performed LC-MS/MS experiments that showed that RutR from Tu et al. is not quantitatively lysine acetylated. Instead overall acetylation occupancy is very low, app. 1%. We found K21 and K150 to be the major acetyl-acceptor lysines from low-occupancy sites. We found K52 in RutR from Tu et al. to be acetylated with occupancy of 0.02% and K62 was not at all found to be acetylated. This shows that the model from Tu et al. cannot be valid. For such a quantitative (auto-proteolytic) degradation of RutR as suggested by Tu and coworkers, if elicited by YcgC mediated deacetylation of RutR AcK52 and AcK62, the lysines have to be also almost quantitatively acetylated.

In the current work, YcgC was expressed as a GST-tagged protein in E. coli, purified, and the tag removed by TEV protease, and C-terminally His-tagged RutR was obtained acetylated at the previously reported Lys52 and Lys62 by stop-codon suppression methods and purified. Unfortunately, in vitro incubation of the putative deacetylase and acetylated substrate did not lead to any activity.If indeed YcgC is not a deacetylase, it will be very important for the community and hence I believe this work is important. However, there could be explanations for the discrepancies other than that the previous study was flawed, and these need to be explored. Both the YcgC enzyme and its substrate appeared to have been prepared differently in this study than in the previous study and apparently with different affinity tags. To truly test whether the previous work is reproducible, the same constructs should be used.

We exchanged proteins with both groups and performed assays with the original protein preparations from Tu et al.. We clearly see that YcgC is no deacetylase. We only observe reduction of molecular size of all RutR proteins upon treatment with YcgC from Tu et al., but not with YcgC S200A from Tu et al. and not with our YcgC. Furthermore, in contrast to the publication of Tu et al., treatment with catalytically active CobB (prepared by us) did not show any proteolytic degradation of no RutR protein analysed, also not of RutR from Tu et al.

We analysed YcgC and YcgC S200A preparations from Tu et al. (and our YcgC preparation) and found several proteases present only in the YcgC from Tu et al. Using our site-specifically acetylated RutR AcK52 and AcK62 proteins we see reduction in molecular size upon treatment with YcgC from Tu et al. However, the degradation band does still show acetylation signal in immunoblotting showing that deacetylation is not the molecular event preceding (auto-)proteolysis as suggested by Tu et al.

The observations in the current study that the protein "behaved similarly on analytical SEC" is not really a measure of "the high quality of proteins". It shows that the proteins behave the same, but says nothing about the "quality". Similarly, I don't think that attaching the His-tag at the C-terminus of YcgC ensures it retains its activity as is written in the second paragraph of the subsection “Expression and purification of YcgC, CobB and RutR”, and identical behavior on SEC says nothing about proper folding, just that they are folded the same. I do not think the authors can rule out that either YcgC or RutR may be improperly folded in the current study, or that the hyper acetylation of RutR in the previous work may have been important.

We rephrased some of the statements as suggested by reviewer 2. For our RutR proteins (RutR, RutR AcK52 and RutR AcK62) we can exclude a hyperacetylation as we see the correct molecular mass by ESI-MS. We performed LC-MS/MS also with the original RutR preparation from Tu et al. and found that the overall acetylation occupancy is low, only app. 1%. Size-exclusion chromatography can give a hint if the oligomeric state is comparable. If acetylation of RutR would mass up the structure, we would see this also by SEC as most likely we would observe an increase of protein elution very early as aggregates in the exclusion volume. Therefore, we think analytical SEC runs can be used for quality control.

I think at minimum, the authors of the current paper should prepare the same form of YcgC and the same form of RutR as in the previous study (same affinity tags and at the same positions). eLife should assist in requesting materials from Tu et al. In the end it is in the interest of both research groups and of the research community that the conclusions of the previous study are tested, but it is not in the interest of anyone if it turns out that the differences in conditions/proteins caused the apparent discrepancy and that YcgC does in fact have deacetylase activity.

As written above, we exchanged all proteins between both labs and we included data with proteins from Tu et al. in this revised manuscript. These data include immunoblottings using all proteins and both antibodies used for deletion. Additionally, we performed LC-MS/MS with YcgC and RutR preparations Tu et al. These data clearly show that YcgC is no deacetylase. RutR from Tu et al. is not quantitatively lysine-acetylated and this per se shows that the molecular model cannot be correct as quantitative autoproteolysis would also request that acetylation is quantitative. Moreover, although AcK52 was found on RutR from Tu et al. the occupancy is very low (app. 0.1%) and AcK62 was not found at all. Our LC-MS/MS data on ycgC show clearly contaminations with several proteases only present in YcgC from Tu et al. but neither in our YcgC preparation nor in YcgC S200A prepared by Tu et al.

When I read the previous study by Tu et al. I could not find if their proteins for their in vitro data were His-tagged at the N- or C-terminus. This is critical information and if it indeed was not in the paper, eLife should request that information from the original authors.I did not think that the section: "YcgC does not show any deacetylase or proteolytic activity" added much if any value. Tu et al. never suggested YcgC was a general protease like trypsin. Instead they clearly proposed that upon deacetylation (by either YcgC or by CobB) RutR undergoes autoproteolysis. They also never suggested YcgC to act on a p53 based peptide. Hence, I did not think that these studies had much merit. The focus should be on YcgC-catalyzed deacetylation of RutR with proteins that are the same as those in the Tu et al. study. If that indeed leads to no activity, then I think it is important to publish this study.

We removed this section from the revised manuscript.

Reviewer #3:

This report by Lammers et al. rebutted the previous discovery by Tu et al. that Ycgc is a unique protein deacetylase in bacteria.Subsection “Expression and purification of YcgC, CobB and RutR”, first paragraph: GSH should be GST.Figure 2A: in this reviewer's visual judgement, YcgC had some deacetylase activity but YcgC S200A did not.Figure 2B was poorly presented and easy to cause confusion. The authors should include data of RutR WT, AcK52 and AcK62 for each cluster (i.e. Control, YcgC, and CobB). Error bars should be included. The figure legend does not seem to match the figure graph.

We corrected the points suggested by reviewer 3, for YcgC, we are sure that it does not have deacetylase activity. If it acted like an enzyme the signal should be completely gone regarding the huge enzyme:substrate ration we used in this assay. These assays are meant only to make qualitative statements. In Figure 2B we show the quantification of ESI-MS spectra of the reactions as described. These reactions are shown in A using immunoblotting as a readout. We corrected it accordingly to make this clearer.

YcgC is a bacterial protein but p53 is a mammalian protein; therefore it is not a surprise that p53 peptide is not a substrate of YcgC. This point is particularly important as the previous study pointed out that YcgC has very narrow substrate specificity.

Reviewer 3 is correct in saying that p52 might not be a substrate as also already mentioned by reviewer 2. Therefore, we deleted this section from the revised manuscript.

Tu et al. showed that RutR expressed in E. coli W3110 was highly acetylated, but this is not the case in this present study where BL21 (DE3) was used. Such drastic difference means physiologies of the two cellular systems are distinct and conclusions on RutR acetylation cannot be compared. This deserves further investigation.

We performed LC-MS/MS analyses on the original RutR protein preparation from Tu et al. We found that lysine acetylation occupancy of the RutR of Tu et al. is very low (app. 1%). We found that K21 and K150 are the major acetyl-acceptor sites, although of course occurring at very low occupancy (0.3% and 0.7%, respectively). We found K52 to be acetylated with about 0.02% and K62 was not identified to be acetylated at all. These data show that the working model presented by Tu et al. cannot be correct.

This reviewer is worried about whether the differences in the protein constructs, acetyl-lysine antibody, and experimental assay protocols used between Tu's study and this one, may be the main reasons accounting for the divergent conclusions on whether YcgC has deacetylase activity. YcgC was assumed to have different mechanism from CobB, but why did the authors used the same protocols for deacetylase activity test with NAD present?

We exchanged proteins with both groups and performed assays with the original protein preparations from Tu et al. under the same conditions. We see clearly that YcgC is no deacetylase. We only observe reduction of molecular size of all RutR proteins upon treatment with YcgC from Tu et al., but not with YcgC S200A from Tu et al. and not with our YcgC. We analysed YcgC and YcgC S200A preparations from Tu et al. (and our YcgC preparation) by LC-MS/MS and found several proteases present only in YcgC preparation by Tu et al. Amongst these proteases we found Lon protease in YcgC by Tu et al. Using our site specifically acetylated RutR acK52 and AcK62 proteins we see reduction in molecular size upon treatment with YcgC preparation from Tu et al. However, the degradation band does still show acetylation signal in immunoblotting showing that deacetylation is not the molecular event preceding (auto-)proteolysis as suggested by Tu et al.. Furthermore, treatment with catalytically active CobB did not show any proteolytic degradation of no RutR protein analysed, also not of RutR from Tu et al. This is also in contrast to the data shown by Tu et al. in the original publication.

[Editors' note: the author responses to the re-review follow.]

In the third paragraph of the Discussion, the authors provide quantification of the amount of acetylation of RutR in the preparation of Tu et al. They use the fact that they achieve 90% sequence coverage as argument that the mass spectrometry intensities can be used for quantification. The reviewers do not think this is valid. If the acetylated peptides ionize much more poorly than the nonacetylated peptides, then peak intensities can only be used if the authors have standard curves of mixtures of different ratios. Hence, they should be more qualitative in their discussions.In Figure 2A, what is in the ctrl lane? Also, do YcgC and CobB happen to have the same mass? If so, it may be good to provide the calculated molecular weights of RutR, YcgG and CobB in the legend to avoid confusion for readers.

We are delighted that the reviewers are now convinced that YcgC has not the desired deacetylase activity. We included all of the reviewers’ and the *eLife* editors’ suggestions in the revised version of the manuscript. From my point of view this strongly improved the manuscript in terms of its conciseness and in its tone.